# *CcNAC1* by Transcriptome Analysis Is Involved in Sudan Grass Secondary Cell Wall Formation as a Positive Regulator

**DOI:** 10.3390/ijms24076149

**Published:** 2023-03-24

**Authors:** Yanzhong Huang, Chen Qian, Jianyu Lin, Augustine Antwi-Boasiako, Juanzi Wu, Zhiwei Liu, Zhengfeng Mao, Xiaoxian Zhong

**Affiliations:** 1National Forage Breeding Innovation Base (JAAS), Key Laboratory for Saline-Alkali Soil Improvement and Utilization (Coastal Saline-Alkali Lands), Ministry of Agriculture and Rural Affairs, Institute of Animal Science, Jiangsu Academy of Agricultural Sciences, Nanjing 210014, China; 2National Center for Soybean Improvement, Key Laboratory of Biology and Genetics and Breeding for Soybean, Ministry of Agriculture, State Key Laboratory of Crop Genetics and Germplasm Enhancement, Nanjing Agricultural University, Nanjing 210095, China; 3Crops Research Institute, Council for Scientific and Industrial Research, Kumasi P.O. Box 3785, Ghana; 4College of Agro-Grassland Science, Nanjing Agricultural University, Nanjing 210095, China

**Keywords:** Sudan grass, secondary cell wall, RNA-seq, *CcNAC1*, protein interaction

## Abstract

Sudan grass is a high-quality forage of sorghum. The degree of lignification of Sudan grass is the main factor affecting its digestibility in ruminants such as cattle and sheep. Almost all lignocellulose in Sudan grass is stored in the secondary cell wall, but the mechanism and synthesis of the secondary cell wall in Sudan grass is still unclear. In order to study the mechanism of secondary cell wall synthesis in Sudan grass, we used an in vitro induction system of Sudan grass secondary cell wall. Through transcriptome sequencing, it was found that the *NAC* transcription factor *CcNAC1* gene was related to the synthesis of the Sudan grass secondary cell wall. This study further generated *CcNAC1* overexpression lines of Arabidopsis to study *CcNAC1* gene function in secondary cell wall synthesis. It was shown that the overexpression of the *CcNAC1* gene can significantly increase lignin content in Arabidopsis lines. Through subcellular localization analysis, *CcNAC1* genes could be expressed in the nucleus of a plant. In addition, we used yeast two-hybrid screening to find 26 proteins interacting with CcNAC1. GO and KEGG analysis showed that CcNAC1 relates to the metabolic pathways and biosynthesis of secondary metabolites. In summary, the synthesis of secondary cell wall of Sudan grass can be regulated by *CcNAC1*.

## 1. Introduction

Sudan grass (*Sorghum sudanense* (Piper) Stapf.) is an annual sorghum plant native to Africa. It is widely adapted and used by all kinds of herbivorous livestock and poultry due to its high biological yield, nutrient profile, palatability, and barren tolerance [1]. The main nutrient in Sudan grass, and other herbage for ruminants, is lignocellulose, in which cellulose and hemicellulose can be digested and absorbed by microorganisms in the rumen of ruminants such as cattle and sheep [2]. Lignin maintains plant mechanical strength, promotes water and nutrient transport, and resists environmental stresses [3]. The majority of lignocellulose in Sudan grass is stored in the plant’s secondary cell wall [4]. Therefore, it is important to study the biosynthesis of the secondary cell wall of Sudan grass to improve its digestibility by livestock while increasing its biological yield.

During plant growth, tissue cell differentiation occurs which results in cells undergoing secondary cell wall thickening processes [5]. The initiation of secondary cell wall synthesis requires coordination between plant development and environmental signals, and the regulation of the expression of cellulose, hemicellulose, and lignin synthesis genes through a complex multistage transcriptional regulatory network [6]. Transcription factors include NAC (*NAM*, ATAF1/2, CUC2) and MYB as core members involved in the synthesis of different cell wall components, processing, and modification of different carbohydrate main and branch chains, and cross-linking and depolymerization between different macromolecules [7].

In plants, two types of NAC domain transcription factors, *VND* (*vascular-related NAC domain*) and *NST/SND* (*NAC secondary wall thickening promoting factor/secondary wall-associated NAC domain protein*), regulate the secondary wall synthesis of vascular duct cells and fiber cells, respectively [8,9]. In Arabidopsis, *AtVND6* can induce epigenous xylem thickening by regulating programmed cell death in conduit cells [10]. *AtVND7* regulates the development of vascular tissues by controlling the expression of *AtREV* (*Revoluta*) and *AtPHB* (*Phabulosa*) genes [11,12]. Overexpression of *AtVND7* in Arabidopsis leads to secondary wall thickening in the protoxylem of the plant [13]. AtVNI2 (VND-interacting2) can inhibit the function of AtVND7 by interacting with AtVND7 [14]. In bananas, *MnVND6/MnVND7* genes are activated by upstreaming *MnSNBE-like* sites to regulate xylem secondary cell wall synthesis [15].

The second type of transcription factor, *AtNST1/AtNST3* (*AtSND1*), is a key factor in regulating the secondary wall thickening of xylem fiber cells [8,9]. The expression of *AtSND1* relates to the thickening of the secondary wall in the fibers, and inhibition of *AtSND1* expression can significantly reduce the secondary wall thickness of the fibers [16]. The expression of synthetic genes such as cellulose, hemicellulose, and lignin were inhibited in *nst1nst3* double mutants. The thickening of other intervascular fibers and xylem secondary cell walls were completely inhibited but the xylem duct was not affected [17].

The *MYB* transcription factor downstream of *NAC* is a secondary regulator of secondary cell wall synthesis and is involved in regulating the synthesis of components of the secondary cell wall [18]. The upstream *NAC* transcription factor activates the expression of 14 downstream genes related to lignin and cellulose synthesis by directly regulating *AtMYB46* and *AtMYB83* [19]. An *myb46myb83* double mutant caused no secondary cell wall formation in the plant leading to plant wilt and death [5]. Similarly, in woody plants such as pine (*Pinus taeda* L.) and eucalyptus (*Eucalyptus grandis* H.), overexpression of homologous genes of *AtMYB46* and *AtMYB83* activate the expression of genes relating to secondary cell wall synthesis, thus, regulating the synthesis of lignin and other substances in plants [20,21]. *MYB* family genes such as *AtMYB20* and *AtMYB42* are also involved in plant secondary cell wall synthesis by controlling lignin and phenylalanine biosynthesis [22,23], whereas *AtMYB52* and *AtMYB54* negatively regulate xylem secondary cell wall thickening [24], while *AtMYB58* and *AtMYB63* activate lignin accumulation in cells [25]. Thus, plants can activate downstreaming of *MYB* secondary transcription factors through *NAC* transcription factors, resulting in regulating the expression of downstream transcription factors and secondary cell wall synthesis genes.

With the development of omics technology, including genomics, transcriptomics, proteomics, and metabolomics, researchers are able to understand the function of genes and proteins in plant physiological process. With the reduction of sequencing cost and upgrading of technology, transcriptome analysis based on RNA-seq has become one of the most effective strategies to analyze differential gene expression before and after treatment application [26]. A total of 62 transcription factor members of the *NAC* (*NAM*, *ATAF1/2* and *CUC2*) family were obtained and seven *PdeNAC* genes were preferentially expressed by wood tissue-specific transcriptome analysis in *Pinus densiflora* (Korean red pine) [27]. In the legume *A. auriculiformis* and *A. mangium*, five transcriptionally regulated *R2R3-MYB* genes are involved in secondary cell wall formation and lignin deposition by sequencing the transcriptome of young stem and inner bark tissue [28]. Overexpression of this gene in Arabidopsis showed that *CcNAC1* can positively regulate plant lignin synthesis. At present, there is no reference genome for Sudan grass, and the basic research is weak and lacks sufficient molecular markers. Therefore, it is difficult to conduct research through a traditional map-based cloning strategy coupled with a waiting period to obtain results. Therefore, the establishment of the in vitro secondary cell wall induction system of Sudan grass combined with transcriptome sequencing is a new and faster way for the biosynthesis of the secondary cell wall of Sudan grass. The functional analysis of the *CcNAC1* gene can provide a theoretical basis for study of the synthesis of the secondary cell wall of Sudan grass and germplasm resources for genetic engineering breeding for forage quality. Our study provides evidence on the function of CcNAC1 protein in Sudan grass and its regulatory pathway.

## 2. Results

### 2.1. Induction of Secondary Cell Wall Establishment by Sudan Grass In Vitro

We found that the callus of Sudan grass does not contain secondary cell walls, but under the induction condition of light and brassinosteroids (BRs) it can produce secondary cell walls. Hence, we established an in vitro induction system of Sudan grass callus secondary cell wall to study its synthesis pathway. We induced the Sudan grass callus to produce secondary cell wall tissue by adding 5 g/L activated carbon and 2 μM BRs to the medium under 16 h/d light treatment (Figure 1A). After 18 days, the production of secondary cell wall reached a plateau (Figure 1B). We then used transcriptome analysis to further explore the synthesis mechanism of Sudan grass secondary cell wall and related genes using the in vitro induced tissue of Sudan grass secondary cell wall.

### 2.2. Transcriptome of Secondary Cell Wall Synthesis of Sudan Grass

Five cDNA libraries induced by light and hormones in Sudan grass were constructed, and the transcriptome was sequenced. A total of 221,936,940 sequence read fragments (reads) were obtained, including 27,739,721,621 base sequences (bp) (Appendix A). The reads were sequence assembled to obtain 67,070 single gene clusters (unigene) (Appendix A). Principal component analysis (PCA) showed that the effect of light conditions on callus gene expression was less than that induced by hormones (Figure 2A). For instance, CK (control), TEMB2C5+Light18day expression profile had 1413 genes up-regulated and 1246 genes down-regulated; TEMB2C5+Dark18day expression profile had 1537 genes up-regulated and 1254 genes down-regulated; TEM+Light18day expression profile had 782 genes up-regulated and 1285 genes down-regulated, and TEM+Dark18day expression profile had 1236 genes up-regulated and 1090 genes down-regulated (Figure 2B and Appendix A).

### 2.3. Functional Classification of Unigene of Secondary Cell Wall Synthesis of Sudan Grass

The effect of BRs on secondary cell wall gene expression was stronger than that of light. To further explore gene responses to secondary cell wall synthesis of Sudan grass, we analyzed unigene functional annotations of callus BRs and dark treatment (TEMB2C5+Dark18day vs. CK). According to GO functional annotation, 64 branches could be roughly divided into three categories: biological process (BP), cellular component (CC) and molecular function (MF) (Figure 3A). Most of the Unigenes were related to the condensin complex, kinesin complex, and the extracellular region. The GO functional annotation for TEM+Dark18day vs. CK, TEM+Light18day vs. CK, and TEMB2C5+Light18day vs. CK are shown in Appendix A. The results from the KEGG analysis of Unigenes showed that their functions can be roughly divided into 20 categories (Figure 3B). A KOG database comparison analysis revealed that the unigenes can be mapped to 112 metabolic pathway branches according to metabolic pathways, such as phenylpropanoid biosynthesis, flavone and flavonoid biosynthesis, fatty acid elongation, flavonoid synthesis, phenylalanine, tyrosine and tryptophan biosynthesis, plant hormone signal transduction, and degradation of aromatic compounds (Appendix A). The KEGG pathway analysis for TEM+Dark18day vs. CK, TEM+Light18day vs. CK, and TEMB2C5+Light18day vs. CK are shown in Appendix A. 

### 2.4. Analysis of CcNACs Family Genes Related to Secondary Cell Wall Synthesis

In the TEMB2C5+Dark18day, TEMB2C5+Light18day and TEM+Light18day treatments, the secondary cell wall in the Sudan grass callus was induced and synthesized. The result shows that 51 genes were upregulated while 180 genes were down regulated in all the three treatments (Figure 4A,B, and Appendix A). In plants, the *NAC* transcription factor family is an upstream regulator of secondary cell wall synthesis, so this study focused on the analysis of *NAC* family transcription factors in transcriptome data. Through analysis, it was found that nine different *CcNACs* transcription factors had common in these three treatments. (Figure 4C). Five of the *CcNACs* genes were up-regulated under light and BRs treatment, *CcNAC1* and *CcNAC2* were not expressed in CK but were up-regulated under light and BRs treatment. The expression level of *CcNAC1* was higher than that of *CcNAC2*. The qRT-PCR data were consistent with the transcriptomic data (Figure 4D). Therefore, *CcNAC1* was selected for further functional verification.

### 2.5. CcNAC1 Increases the Lignin Content of Plants

We generated *CcNAC1* overexpressing Arabidopsis lines (OE1 and OE2) to investigate the function of *CcNAC1* in secondary cell wall synthesis. The overexpressing of *CcNAC1* lines (OE1 and OE2) were significantly lower than that of the wild type (WT) (Figure 5A). The overexpression lines and the WT were sampled for their lignin content. The results showed that the lignin contents of the overexpression lines were significantly higher (2.2 times) than that of the WT (2 times) (Figure 5B).

### 2.6. CcNAC1 Encodes a Nuclear Localization Protein

To investigate the mechanism of CcNAC1 protein regulation in plant secondary cell wall synthesis we performed subcellular localization of the functional regions of CcNAC1. Online prediction of the NAC protein domain showed that CcNAC1 was encoded by 341 amino acids, among which 6-130 amino acids were the NAM domain (Figure 6A). It is predicted that CcNAC1 is a nuclear localization protein.

To verify the online prediction results, we constructed the CcNAC1 and EGFP fusion expression vector *35S-CcNAC1-EGFP*, which controlled by the *35S* promoter (Figure 6B). Transient expression of fusion expression vectors and empty vectors in *N. benthamiana* leaves was performed to assess CcNAC1 subcellular localization. DAPI was also used as a nuclear marker. The green fluorescent protein (GFP) signal was observed through laser confocal microscopy. The results showed that a GFP signal of an empty vector was detected in the cytoplasm, cell membrane, and nucleus (Figure 6F–H). In contrast, the fusion protein expressed by *35S-CcNAC1-EGFP* was only detected in the nucleus (Figure 6C–E). These results demonstrate that CcNAC1 is localized in the nucleus.

### 2.7. Candidate Interaction Proteins Analysis of CcNAC1

Using NAC as the bait protein to screen the yeast library of Sudan grass secondary cell wall and analyze the molecular biological function of CcNAC1 interacting protein indicated the regulatory pathway of CcNAC1 in the synthesis of Sudan grass secondary cell wall. Through GO annotation, 26 proteins interact with CcNAC1 in secondary cell wall synthesis, and interacting proteins were found to be associated with one or more of the following three GO categories: Biological Process (BP), Cellular Component (CC) and Molecular (Figure 7A). Analysis of the biological processes indicated that the interaction proteins were mainly involved in processes relating to cellular (GO:0009987), metabolic (GO:0008152), single-organism (GO:0044699), biological regulation (GO:0065007), and regulation of biological (GO:0050789), among other processes. Analysis of the cell composition indicated that the interaction proteins were mainly parts the cell (GO:0005623), cell components (GO:0044464), membrane (GO:0016020), organelle (GO:0043226) and membrane components (GO:0044425). Furthermore, a molecular biological function analysis demonstrated that proteins interactions mainly included structural components of binding (GO:0005488), catalytic activity (GO:0003824), and transporter activity (GO:0005215), among others (Appendix A).

KEGG enrichment analysis helps us to understand the metabolic pathways involved in genes. We performed KEGG enrichment analysis to increase our understanding of gene metabolic pathways in secondary cell wall synthesis in plants. The KEGG enrichment analysis of the 26 candidate interaction proteins of CcNAC1 indicated that the proteins mainly involve 12 pathways (Figure 7B), including metabolic pathways (ko01100), biosynthesis of secondary metabolites (ko01110), plant hormone signal transduction (ko04075), and other regulatory pathways (Appendix A).

## 3. Discussion

With improved living standards, demand for animal husbandry products is on the rise. Increasing the fiber content of forage grass is one of the ways to increase the utilization rate of feed for ruminants such as cattle and sheep in order for them to yield high quality products. Sudan grass callus does not contain secondary cell walls under dark conditions, but can produce secondary cell walls under the induction of light and BRs. Therefore, we performed transcriptome sequencing to analyze differentially expressed secondary cell wall regulation to switch on the *NAC* family during secondary cell wall synthesis [29]. It was found that *CcNAC1* was expressed along with the synthesis of the secondary cell wall of Sudan grass. Through functional verification, it was found that *CcNAC1* was a positive regulator of the biosynthesis of the secondary cell wall of Sudan grass. Light can induce changes in the structure and play a role in the function of plant xylem and promote the growth of plant secondary cell walls [30]. For instance, in cotton, under shade conditions, the number of vascular bundles and the degree of lignification were decreased [31]. In poplar, dark treatment resulted in thinning of poplar secondary cell walls and reduced the density of the wood [32]. Studies have shown that blue-violet light in sunlight can induce the expression of the *AtCRY1* gene and promote the expression of *AtMYC2/AtMYC4* [33]. *AtMYC2/AtMYC4* can directly bind to the *AtNST1* promoter and activate its expression, thereby promoting the synthesis and thickening of secondary cell walls [34]. *AtMYB46* regulates secondary cell wall synthesis genes (*AtCesA4* (*cellulose synthase 4*), *AtCesA7*, and *AtCesA8*) under the induction of light [35].

BRs play an important role in plant growth and development [36]. Studies have shown that BRs are related to the development of plant xylem [37], and there is a synergistic association between the synthesis of BRs in plants and the development of plant xylem [38]. In binding of BRs to the receptor BRI1 (BRASSINOSTEROID INSENSITIVE 1), SERK3 (BAK1/somatic embryogenesis receptor kinase) is phosphorylated, thereby initiating a signaling cascade reaction. BRs signaling could induce the expression of BZR1 and BES1/BZR2, thereby regulating the gene expression of cell wall biosynthesis such as PMEs [39]. Deficiency in BRs synthesis in plants leads to dwarfism in plant phenotypes, with decrease in lignin and cellulose content [40]. For instance, in poplar (*Populus* L.), knockout of the BRs receptor by gene editing inhibits the differentiation of secondary cell walls in the xylem, resulting in marked plant dwarf and growth-deficient phenotypes [41]. Studies have shown that in the presence of BRs, the transcription factor *AtBZR1* of the BRs pathway binds to the cellulose and enzyme gene *AtCesA6* [42], and the *AtBES1* transcription factor can combine with other *AtCesA* family genes to jointly regulate the synthesis of plant secondary cell walls [43]. Therefore, natural light and BRs can induce the synthesis of secondary cell walls in plants. In this study, Sudan grass callus treated with light and BRs showed that BRs induce secondary cell walls formation more than that of light (Figure 2A). Therefore, the ability of BRs to induce the secondary cell wall of Sudan grass callus is stronger than that of light.

In the KEGG pathway analysis, the differential genes were mainly concentrated in the phenylalanine metabolism pathway, flavonoid synthesis, plant-pathogen interaction, plant hormone biosynthesis and signal transduction (Figure 3B). Both lignin and flavonoids belong to a group of phenylpropanoids that share Coumaroyl-CoA in two metabolic pathways [44]. Visible and UV light could activate the expression of flavonoid biosynthetic genes by regulating the *R2R3-MYB* gene [45]. Moreover, the gene expression level of flavonoid biosynthesis genes is positively correlated with light intensity, but too strong light inhibits the expression of related genes [46]. It could be speculated that the synthesis of secondary cell walls of Sudan grass under light conditions might be related to the biosynthesis pathway of flavonoids induced by light. During the synthetic processes of plant cells, phenylalanine is synthesized into lignin through a series of catalytic reactions under the catalysis of peroxidase [47]. Previous studies have shown that the key enzymes *AtADT4* and *AtADT5* in the synthesis of flavonoid anthocyanins are related to the synthesis of secondary cell walls. The mutants *AtADT4* and *AtADT5* resulted in softening of plant stems and phenotypes that could not provide support [48]. When a pathogen invades plants, the secondary cell wall is thickened at the invasion site to block further invasion by the pathogen [49]. In these pathways, plant hormones are involved and play a very important regulatory role [50,51].

The discovery of key regulatory genes for secondary cell wall synthesis in Sudan grass requires the search for genes with common expression trends in treatments with secondary cell wall synthesis. It was clear from the Wenn Diagram that 51 genes were up-regulated and 180 genes were down-regulated in the three secondary cell wall synthesis treatments (TEMB2C5+Dark18day, TEMB2C5+Light18day and TEM+Light18day) (Figure 4A,B, and Appendix A). Therefore, these 231 genes might be involved in regulating secondary cell wall synthesis in Sudan grass. The NAC transcription factors play a very important role in the regulation of secondary cell wall synthesis in plants as a primary transcription factor [8,9]. Ours results on the differentially expressed NAC transcription factors and the expression trend of *CcNAC1* and *CcNAC2* genes were consistent with the synthesis of secondary cell wall in Sudan grass callus (Figure 4C). Therefore, the expression of these two genes may regulate the synthesis of the secondary cell wall of Sudan grass. The expression of the *CcNAC1* gene was higher than that of *CcNAC2*. So, *CcNAC1* was selected for validation. We overexpressed the *CcNAC1* gene in Arabidopsis through heterologous expression. The lignin content of 2-month-old plants with *CcNAC1* overexpressed in Arabidopsis lines (OE1 and OE2) was significantly higher than that of wild-type plants (Figure 5B), suggesting that the *CcNAC1* gene is a positive regulator of secondary cell wall synthesis. The two *CcNAC1* overexpression lines (OE1 and OE2) showed dwarf phenotype (Figure 5A). Studies have shown that lignin content directly affects plant growth and development. Plant root development is negatively correlated with lignin content, and excessive lignin accumulation inhibits the development and elongation of roots in the case of Arabidopsis and carrot. Treatment of plants with exogenous IBA resulted in decreased lignin accumulation and increased root length [52]. Lignin content not only affects root development, but also affects fruit size. In peanuts (*Arachis hypogaea* L.), plants with high lignin content also have a relatively small pod [53]. This means that the dwarf phenotype of plants caused by high lignin content may be caused by the inability of cells to elongate due to excessive lignin in cells.

GO annotation found that some interaction proteins relate to *CcNAC1* functions involving in cellular process and cell (Figure 7A). Secondary cell wall synthesis is part of plant morphogenesis. The enrichment of *CcNAC1* interacting proteins and cell progression suggest that *CcNAC1* and its interacting proteins are involved in the regulation of cell morphogenesis [54]. KEGG analysis showed that *CcNAC1* interacting proteins were mainly enriched in metabolic pathways and biosynthesis of secondary metabolites, indicating that *CcNAC1* interacting proteins are mainly related to the synthesis of secondary metabolites (Figure 7B). This indicate that NAC interacting proteins involved in the pathway may also be related to secondary cell wall formation [55]. Therefore, *CcNAC1* positively regulates secondary cell wall synthesis in plants by regulating the synthesis of related secondary metabolites and participating in cell processes.

## 4. Materials and Methods

### 4.1. Plant Materials and Growth Conditions

Sudan grass Sumu No.3 was used as the plant material for different experiments. Sudan grass cultivar Sumu NO.3 and Arabidopsis cultivar Columbia-0 were used in the laboratory. Secondary cell wall induction was carried out in a controlled growth chamber at 25 °C for 18 days from Sudan grass callus that had been subcultured twice. TEM+Light18 day represented Sudan grass callus cultured in the subculture medium (MS basic salt, B5 vitamin, sucrose 30 g/L, agar 8g/L, 2,4-D 2 mg/L and 6-BA 0.05 mg/L, pH = 5.8) and placed in a temperature at 25 °C under long day (16/8 h light/dark) conditions for 18 days. TEM+Dark18 day represented Sudan grass callus cultured in the subculture medium and placed in a temperature at 25 °C in the absence of light conditions for 18 days. TEMB2C5+Light18 day represented Sudan grass callus cultured in the subculture medium supplemented with 5 g/L activated carbon and 2 μM brassinolide (BRs) (MS basic salt, B5 vitamin, sucrose 30 g/L, agar 8g/L, 2,4-D 2 mg/L, 6-BA 0.05 mg/L, activated carbon 5g/L and BRs 0.96 mg/L, pH = 5.8), and placed in a temperature at 25 °C under long day (16/8 h light/dark) conditions for 18 days. TEMB2C5+Dark18 day represented Sudan grass callus cultured in the subculture medium supplemented with 5 g/L activated carbon and 2 μM BRs, and placed in a temperature at 25 °C in the absence of light conditions for 18 day. CK was treated on the 0 day in the subculture medium of Sudan grass callus culture.

### 4.2. Lignin Dyeing and Content Measurement Method

Phloroglucinol staining was used to stain secondary cell walls of callus of Sudan grass. The Sudan callus was placed in a 10mL centrifuge tube, and 6 M HCl was added. After 30 s, the same amount of phloroglucinol dye solution (phloroglucinol 5 g, and 95% alcohol 100 mL) was added and mixed. The callus was stained for 2 min, and then photographed.

The lignin content of different Arabidopsis lines was determined using a lignin assay kit. Wild-type (WT) lines and *CcNAC1* overexpression lines (OE1 and OE2) were grown under normal growth conditions for 2 months. Then, lignin content was determined using a lignin content assay kit (AKSU010U) following the manufacturer’s protocol (Boxbio, Beijing, China). Three independent replicates were assayed and their standard errors (SEs) calculated.

### 4.3. Phenylalanine Ammonia-Lyase (PAL) Activity Measurement Method

The PAL activity of Sudan grass callus was determined using a PAL activity assay kit. In a controlled growth chamber at 25 °C, the callus of Sudan grass subcultured twice was induced by light with 5 g/L activated carbon and 2 μM BRs treatment (TEMB2C5+Light) and dark treatment (TEM+Dark). Samples were taken from the two treatments every 3 days. PAL activity of samples was determined using a PAL activity assay kit (BC0215) following the manufacturer’s protocol (Solarbio, Beijing, China). Three independent replicates were assayed, and their standard errors (SEs) calculated.

### 4.4. Transcriptome Sequencing and Analyses

Total RNA of Sudan grass callus was extracted using an mirVana miRNA Isolation Kit (Ambion, Austin, TX, USA). All the RNA samples were pooled to construct a complementary DNA (cDNA) library (Integrity Number (RIN) ≥ 7). The libraries were sequenced on an Illumina Hiseq platform (Illumina HiSeq X Ten) and 125 bp/150 bp paired-end reads were generated. Raw data (raw reads) were processed using Trimmomatic. The reads containing ploy-N and the low-quality reads were removed to obtain clean reads. The clean reads were mapped to SWISSPROT database (http://www.uniprot.org (accessed on 4 February 2016)) location information. The library construction and sequencing were carried out at OE Biotech (Shanghai, China)

In transcriptome sequencing analysis, the fragments per kb per million fragments (FPKM) value was applied to represent the expression level of differentially expressed genes under different samples. Principal component analysis (PCA) of gene expression was used to investigate the distribution of samples, to explore the relationship between samples and to verify the experimental design. Genes with an adjusted *p*-value < 0.05 and an absolute value of |log2FoldChange| > 1 were assigned as differentially expressed [26].

### 4.5. Functional Classification of DEGs

Gene Ontology (GO) annotation was obtained from the GO database (https://github.com/tanghaibao/GOatools (accessed on 15 March 2017 and 20 July 2022)). The GO annotation method used the Fisher’s exact test. The GO annotation used four methods, namely Bonferroni, Holm, Sidak, and the false discovery rate, to correct the *p*-value. The enrichment of GO function was considered significant when then *p*-value was lower than 0.05. The Kyoto Encyclopedia of Genes and Genomes (KEGG) database was used for the systematic examination of gene roles, contact genomics, and functional information. KEGG software was used, and KOBAS 3.0 (http://kobas.cbi.pku.edu.cn/home.do (accessed on 16 March 2017 and 21 July 2022)). The Benjamini–Hochberg (BH) method was used to correct the *p*-value. A *p*-value of 0.05 as the threshold, indicated significant enrichment [26].

### 4.6. Quantitative Real-Time PCR (qRT–PCR) Verification

Total RNA from Sudan grass and Arabidopsis was extracted using the RNAprep Pure Plant Kit (Tiangen, Beijing, China), and subjected to reverse transcription using PrimeScript™ RT reagent Kit with gDNA Eraser (Perfect Real Time) (TaKaRa, Kyoto, Japan). Gene-specific primers for qRT-PCR were designed in the GenScript Real-time PCR Primer Design (https://www.genscript.com/tools/real-time-pcr-taqman-primer-designtool), which was accessed on 1 June 2018, and are listed in Appendix A. Quantitative real-time PCR (qRT-PCR) was performed for each cDNA template using ChamQ Universal SYBR qPCR Master Mix (Vazyme, Nanjing, China) with a standard protocol, while the Sudan grass reference gene *CcEIF4a* served as the internal reference gene. Three independent biological replicates were performed for each sample to ensure statistical credibility. The specificity of the reactions was verified by melting curve analysis. The relative mRNA level for each gene was calculated as follows: ratio = 2^−ΔΔCt^ = 2^−[Ct,t-Ct,r]^ (C_t_ cycle threshold, C_t,t_ Ct of the target gene, C_t,r_ C_t_ of the control gene). These reactions were conducted using an Applied Biosystems™QuantStudio™5 Real-Time PCR instrument [26].

### 4.7. Subcellular Localisation of CcNAC1 Protein

The protein domain of CcNAC1, using its amino acid sequence, was predicted through the online software SMART (http://smart.embl-heidelberg.de/smart/set_mode.cgi?NORMAL=1 (accessed on 20 September 2019)). To verify the localisation of the CcNAC1 protein in plant cells, coding CDS sequences of *CcNAC1* were introduced into *pBinGFP4*. The *pBinGFP4-CcNAC1* and *pBinGFP4* vectors were transformed into Agrobacterium GV3101 strains, and both vectors were transformed into *N. benthamiana* leaves using the *N. benthamiana* leaf transient transformation method. The fluorescence of GFP was detected at 488 nm and 405 nm by an upright confocal microscope (Carl Zeiss, Thornwood, NY, USA) at 48–72 h post-inoculation. The excitation wavelengths used were 488 nm for Enhanced Green Fluorescent Protein (EGFP), and 405 nm for 2-(4-Amidinophenyl)-6-indolecarbamidine dihydrochloride (DAPI).

### 4.8. CcNAC1 Overexpression in Arabidopsis

The *Arabidopsis thaliana* cultivar, Columbia-0, was used as the plant material in gene function verification experiments. Arabidopsis seeds were cultivated in soil after being vernalized at 4 °C for 5 days, and then grown in a controlled growth chamber at a temperature of 22 °C under long day (16/8 h light/dark) conditions and a relative humidity of 70%. Arabidopsis transformations were performed at the time of full flowering.

The full-length CDS of the *CcNAC1* gene was cloned into the *pTF101.1* vector under the control of the *CaMV 35S* promoter (*35S-CcNAC1)*. The primers utilized are listed in Appendix A. The *35S-CcNAC1* construct was transformed in Arabidopsis cultivar Columbia-0 using *Agrobacterium tumefaciens* strain EHA101. The selection marker gene (*bar*) and a *35S* promoter were amplified by PCR to verify the positive transgenic plants. Arabidopsis plants were transformed by the floral dip method. The seeds harvested after infection were T_1_ generation. T_1_ seeds were collected and selected on MS medium containing 20 mg/L Basta and then tested *bar*, *CcNAC1* and *35S* promote by PCR [56]. The seeds harvested from positive plants of T_1_ generation were T_2_ generation, and T_2_ generation plants were identified as positive plants by the same screening method as T_1_ generation plants. If all T_2_ generation plants of a line were positive, the line was considered to be homozygous; if separation occurred, the line was considered to be heterozygous. The seeds harvested from T_2_ homozygous lines were T_3_ homozygous lines. T_3_ homozygous lines were used for further study. Samples of Arabidopsis overexpressed lines and the wild type were taken to determine lignin content when each Arabidopsis line was 2 months old.

### 4.9. Yeast Two-Hybrid Assay

We used the yeast two-hybrid (Y2H) system (Takara, Kyoto, Japan) to screen the yeast library of Sudan grass secondary cell walls, The full-length CDS of *CcNAC1* was amplified and cloned into a *pGBKT7* vector, yielding *pGBKT7-CcNAC1* as bait, while using the Sudan grass callus yeast library plasmids as prey. Then, *pGBKT7- CcNAC1* construct and library plasmids were co-transformed into the Y2H Gold yeast strain. Subsequently, these yeast cells were subsequently cultured on SD/-Trp/-Leu plates with 50 µL to calculate transformation efficiency, and others were subsequently cultured on SD/-Trp/-Leu/-Ade/-His+AbA+50 mM 3-AT plates for interaction protein screening. A single yeast clone was picked and grown on the SD/-Trp/-Leu/-Ade/-His+AbA+50 mM 3-AT plates and transferred to an SD/-Trp/-Leu/-Ade/-His+AbA+50 mM 3-AT+X-α-Gal plate for further screening. The blue yeast single clone interacted with proteins of CcNAC1.

The primers and sequences used in this study were from Genscript (China).

## 5. Conclusions

In this study, the in vitro induction system of the secondary cell wall of Sudan grass was established by treating the callus of Sudan grass with light and brassinosteroids (BRs). Transcriptome analysis of five callus samples with different treatments found that the expression trend of CcNAC1 was consistent with the synthesis of secondary cell walls in Sudan grass callus. Therefore, this gene was selected for further study. Furthermore, using the yeast two-hybrid (Y2H) method, we found that CcNAC1 interacted with many functional proteins relating to WRKY transcription factors, amino acid transporters, and cytochrome p450, among others. Thus, our findings suggest that CcNAC1 is a key regulator of secondary cell wall synthesis in Sudan grass.

## Figures and Tables

**Figure 1 ijms-24-06149-f001:**
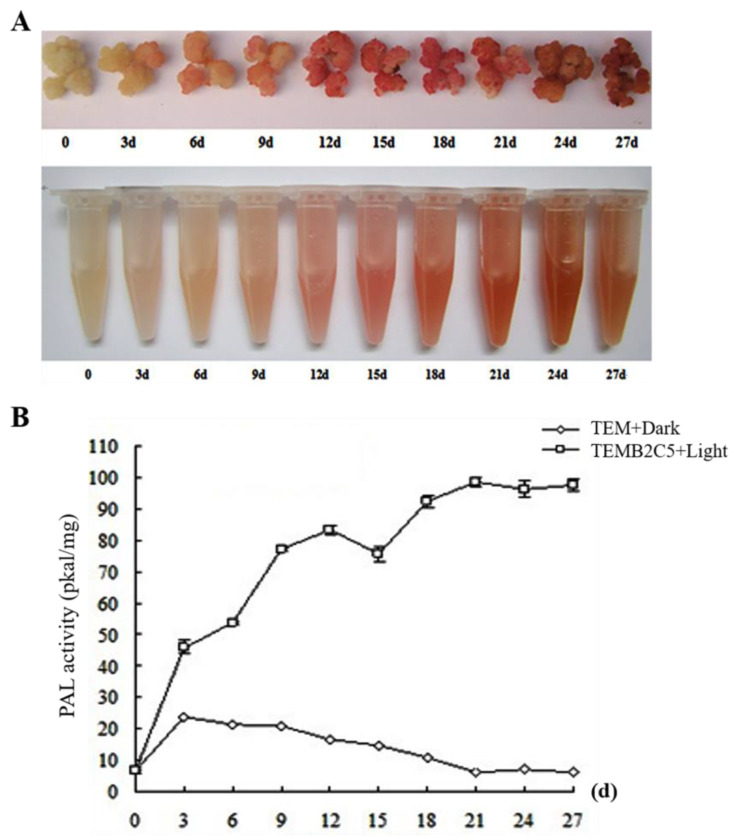
Secondary cell wall induction system of Sudan grass. (**A**) Specific staining of phloroglucinol lignin after callus induction with 2 μM BRs. (**B**) Determination of PAL enzyme activity related to lignin synthesis in callus.

**Figure 2 ijms-24-06149-f002:**
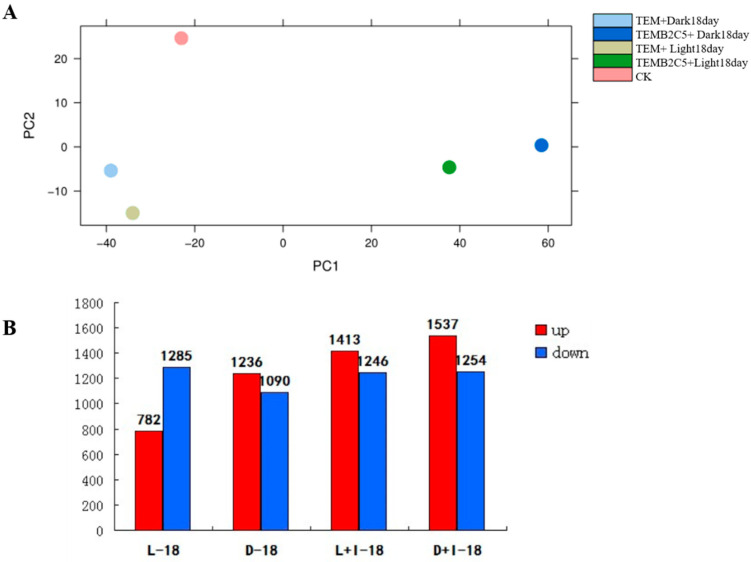
Differentially expressed genes (Unigene) between tested samples. (**A**) Principal component analysis (PCA) of Unigene expression. (**B**) Number of Unigenes compared between treated sample and CK sample are shown in red (up-regulated) and blue (down-regulated).

**Figure 3 ijms-24-06149-f003:**
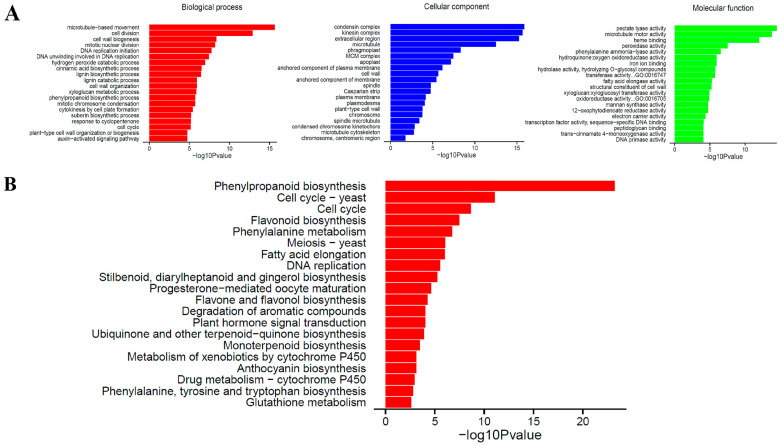
GO and KEGG analysis of unigenes of TEMB2C5+Dark18day and CK differentially expressed. (**A**) Diagram showing GO enrichment analysis of differentially expressed Unigene in two samples. BP, biological processes; CC, cellular components; MF, molecular function. (**B**) KEGG analysis of differentially expressed Unigene in two samples.

**Figure 4 ijms-24-06149-f004:**
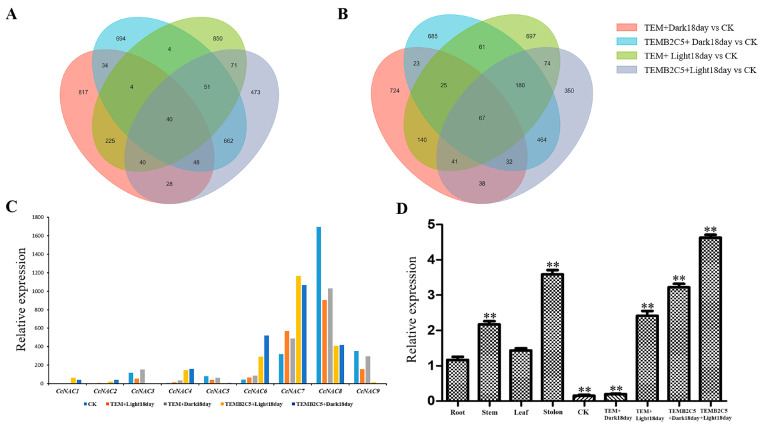
Identification of key *CcNACs* genes for secondary cell wall synthesis biosynthesis in Sudan grass. (**A**) Venn diagram of the up-regulated expression by Unigene in four treatments. (**B**) Venn diagram of the down-regulated expression by Unigene in four treatments. (**C**) Analysis of the expression levels of nine different *CcNACs* genes among five expression profiles. (**D**) Analysis of tissue expression specificity of *CcNAC1* gene by qRT-PCR (*n* = 3). The data represent the means ± SEs. **, *p* < 0.01 (Student’s *t* test).

**Figure 5 ijms-24-06149-f005:**
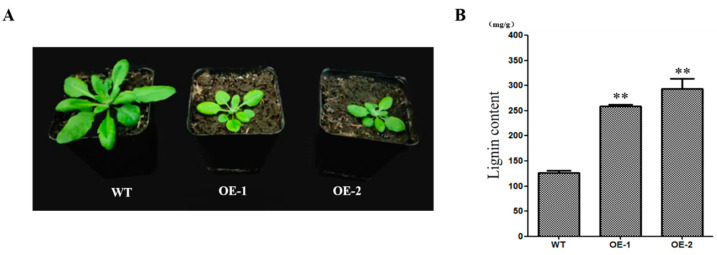
*CcNAC1* positively regulates secondary cell wall synthesis. (**A**) Performance of wild-type (WT) plants and *CcNAC1* overexpression lines (OE1 and OE2) under normal condition. (**B**) Lignin content of wild-type plants and *CcNAC1* overexpression lines 2 months after planting (*n* = 3); Over 30 plants in each line were used for survival rate analysis. The data represent the means ± SEs. **, *p* < 0.01 (Student’s *t* test).

**Figure 6 ijms-24-06149-f006:**
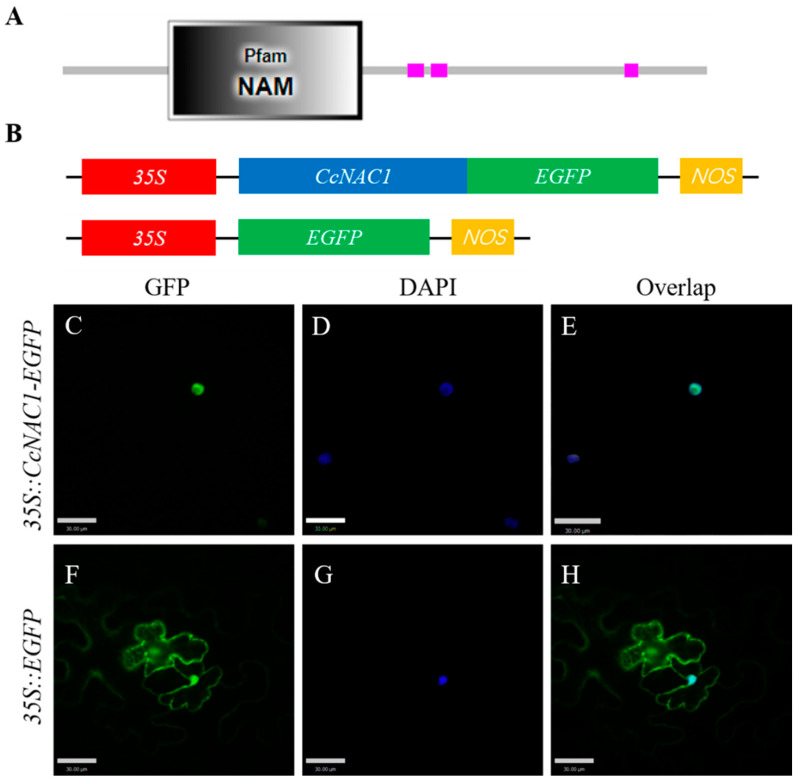
Subcellular localization of CcNAC1 in tobacco cells. (**A**) Amino acid domain prediction of CcNAC1 protein. (**B**) CcNAC1 subcellular localization vector structure. (**C**) *35S-CcNAC1-EGFP* fusion protein. (**D**) DAPI fluorescent dye. (**E**) Merged image of **C** and **D**. (**F**) *35S-EGFP* fusion protein. (**G**) DAPI fluorescent dye. (**H**) Merged image of **F** and **G**. Scale bars are 50 μm.

**Figure 7 ijms-24-06149-f007:**
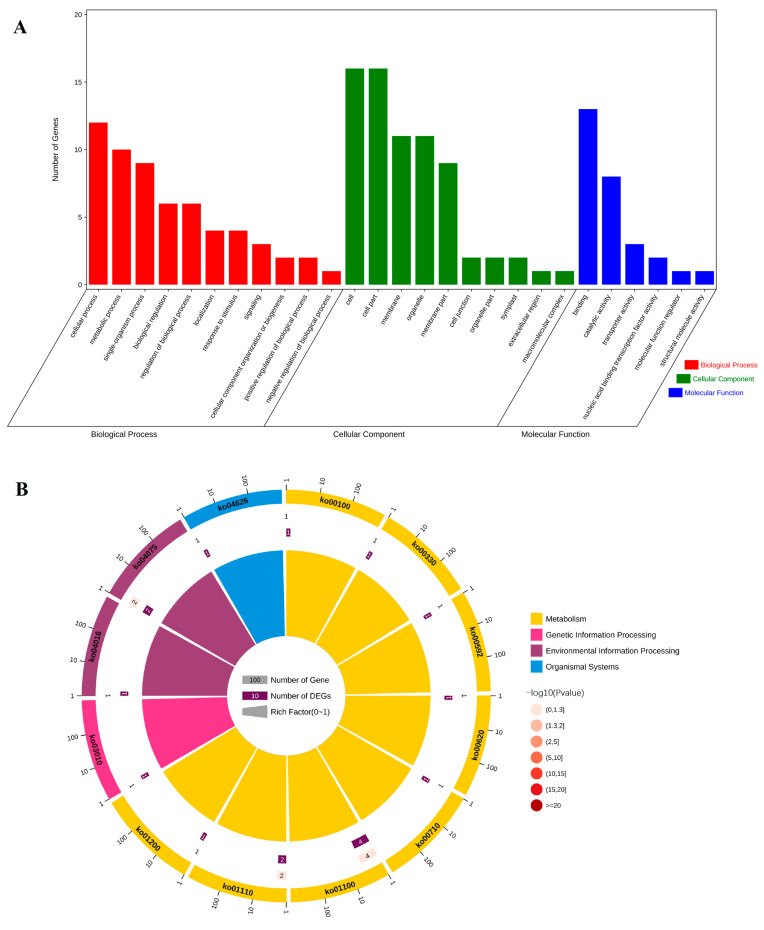
GO and KEGG analysis of CcNAC1 candidate interacting proteins associated with secondary cell wall synthesis. (**A**) Diagram showing GO enrichment analysis of CcNAC1 candidate interacting proteins associated with secondary cell wall synthesis. BP, biological processes; CC, cellular components; MF, molecular function. (**B**) KEGG analysis of CcNAC1 candidate interacting proteins associated with secondary cell wall synthesis.

## Data Availability

The data that support the findings of this study are available from the corresponding author upon reasonable request.

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
