# Peer review of "CcNAC1 by Transcriptome Analysis Is Involved in Sudan Grass Secondary Cell Wall Formation as a Positive Regulator"

_ijms, 2023, doi:10.3390/ijms24076149_

Round 1

Reviewer 1 Report

Dear Authors

Undoubtedly, lignification and the mechanism of the genes and proteins that are involved in the cell wall formation are very important. There are some comments and corrections below that needs to be address;

The role of CcNAC1 gene in secondary cell wall formation is already known (Zhang et al, 2014; DOI: 10.1016/j.gene.2014.05.011). In this study Authors not only identified those genes (CcNAC) by transcriptomic data analysis but also analysed the function and localisation of the CcNAC1 gene as upregulated gene. In the title of the manuscript, more than focusing on the identification of CcNAC1 by transcriptomic data as a gene involved in lignification (which is already known) You should focus on the function part which is novelty of your manuscript. The title of the manuscript doesn’t reflect that.

Line 22-24: Please make it clear that transgenic plants were Arabidopsis.

Line 33: Sorghum sudanense should be italic. In entire manuscript all the Arabidopsis thaliana and other scientific names should be italic and you don’t need to write Arabidopsis thaliana all the time, only Arabidopsis would be enough.

Line 99: give the reference.

In the M&M

Line 344: Cultivar doesn’t need to be italic.

Line 366: what is BRs write the full name please. And a bit more explain about the BR treatment.

Line 440: The title is: “4.9. CcNAC1 Overexpression in Sudan grass Plants” but you overexpressed in Arabidopsis. Correct the title and the line 344 to 350 can be written under the 4.9 subtitle. The method part is a bit messy. In each part you wrote a bit about Arabidopsis. It would be nice all about Arabidopsis such as cultivars, conditions of growth, transformation and so on collect under one subtitle like:  “4.9. CcNAC1 Overexpression” and then explain that you use Arabidopsis the method and so on.

Best wishes

Author Response

Dear reviewer,

We appreciate the critical comments and useful suggestions from you. All the questions and comments have been acknowledged. The manuscript has been also rewritten and organized as suggestions. Herein we re-submit our manuscript entitled “Sudan Grass Transcriptome Analysis Identified CcNAC1 Gene to Function in Secondary Cell Wall Formation” (Manuscript ID: ijms- 2228977) for consideration as article by International Journal of Molecular Sciences.

Sincerely,

Dr. Xiaoxian Zhong

Reviewer: Comments on the manuscript " Sudan Grass Transcriptome Analysis Identified CcNAC1 Gene to Function in Secondary Cell Wall Formation ", by Huang et al.

  1. The role of CcNAC1 gene in secondary cell wall formation is already known (Zhang et al, 2014; DOI: 10.1016/j.gene.2014.05.011). In this study Authors not only identified those genes (CcNAC) by transcriptomic data analysis but also analysed the function and localisation of the CcNAC1 gene as upregulated gene. In the title of the manuscript, more than focusing on the identification of CcNAC1 by transcriptomic data as a gene involved in lignification (which is already known) You should focus on the function part which is novelty of your manuscript. The title of the manuscript doesn’t reflect that.

Re: Thank you. We have modified the title of “CcNAC1 by transcriptome analysis is involved in Sudan grass secondary cell wall formation as a positive regulator” according to your suggestion.

  1. Line 22-24: Please make it clear that transgenic plants were Arabidopsis.

Re: Thank you. We have modified this sentence according to your suggestion.

“This study further generated CcNAC1 overexpression lines of Arabidopsis to study CcNAC1 gene function in secondary cell wall synthesis. It was shown that, the overexpression of CcNAC1 can significantly increase the lignin content in in Arabidopsis lines.”.

  1. Line 33: Sorghum sudanense should be italic. In entire manuscript all the Arabidopsis thaliana and other scientific names should be italic and you don’t need to write Arabidopsis thaliana all the time, only Arabidopsis would be enough.

Re: Thank you. We have updated the scientific name of Sudan grass, checked the scientific name of the entire manuscript.

Sudan grass (Sorghum sudanense (Piper) Stapf.)”.

  1. Line 99: give the reference.

In the M&M

Re: Sorry, we didn't see “In the M&M” on line 99.

  1. Line 344: Cultivar doesn’t need to be italic.

Re: Thank you. We have changed the italic “Cultivar”.

“The Arabidopsis thaliana cultivar, Columbia-0, was used as the plant material in gene function verification experiments.”

  1. Line 366: what is BRs write the full name please. And a bit more explain about the BR treatment.

Re: Thank you. We have added full name “brassinolide” of BRs to the Materials and Methods, and the method of BRs processing is explained in detail.

“Secondary cell wall induction was carried out in a controlled growth chamber at 25°C for 18 days from the Sudan grass callus that had been subcultured twice. TEM+Light18day was the Sudan grass callus cultured in the subculture medium (MS basic salt, B5 vitamin, sucrose 30g/L, agar 8g/L, 2,4-D 2mg/L and 6-BA 0.05mg/L, pH=5.8) and placed in a temperature 25°C under long day (16/8 h light/dark) conditions for 18 days; TEM+Dark18day was the Sudan grass callus cultured in the subculture medium and placed in a temperature 25°C absence of light conditions for 18 days; TEMB2C5+Light18day was Sudan grass callus cultured in the subculture medium supplemented with 5g/L activated carbon and 2 μM brassinolide (BRs) (MS basic salt, B5 vitamin, sucrose 30g/L, agar 8g/L, 2,4-D 2mg/L, 6-BA 0.05mg/L, activated carbon 5g/L and BRs 0.96 mg/L, pH=5.8), and placed in a temperature 25°C under long day (16/8 h light/dark) conditions for 18 days; TEMB2C5+Dark18day was Sudan grass callus cultured in the subculture medium supplemented with 5g/L activated carbon and 2 μM BRs, and placed in a temperature 25°C absence of light conditions for 18 day; CK was treated on the 0 day in the subculture medium of Sudan grass callus culture.”

  1. Line 440: The title is: “4.9. CcNAC1 Overexpression in Sudan grass Plants” but you overexpressed in Arabidopsis. Correct the title and the line 344 to 350 can be written under the 4.9 subtitle. The method part is a bit messy. In each part you wrote a bit about Arabidopsis. It would be nice all about Arabidopsis such as cultivars, conditions of growth, transformation and so on collect under one subtitle like: “4.9. CcNAC1 Overexpression” and then explain that you use Arabidopsis the method and so on.

Re: Thank you. We have modified the title and content in 4.9.

“4.9. CcNAC1 Overexpression in Arabidopsis

The Arabidopsis thaliana cultivar, Columbia-0, was used as the plant material in gene function verification experiments. Arabidopsis seeds were cultivated in the soil, first vernalized at 4°C for 5 days, and then grown in a controlled growth chamber at a temperature of 22°C under long day (16/8 h light/dark) conditions and a relative humidity of 70%. Arabidopsis transformations were performed at the time of full flowering.

The full-length CDS of the CcNAC1 gene was cloned into the pTF101.1 vector under the control of the CaMV 35S promoter. After the vector sequence was confirmed by sequencing, the recombinant pTF101.1-CcNAC1 plasmid vector was transformed into Arabidopsis thaliana cultivar Columbia-0 using Agrobacterium tumefaciens strain EHA101. The selection marker gene (bar) and a 35S promoter were amplified by PCR to verify the positive transgenic plants. Arabidopsis plants were transformed by the floral dip method. The seeds harvested after infection were T1 generation. T1 seeds were collected and selected on MS medium containing 20 mg/L Basta and then tested bar, CcNAC1 and 35S promote by PCR [53]]. The seeds harvested from positive plants of T1 generation were T2 generation, and T2 generation plants were identified as positive plants by the same screening method as T1 generation plants. If all T2 generation plants of a line were positive, the line was considered to be homozygous; if separation occurred, the line was considered to be heterozygous. The seeds harvested from T2 homozygous lines were T3 homozygous lines. T3 homozygous lines were used for further study. Samples of Arabidopsis overexpressed lines and wild type were taken to determine lignin content when each Arabidopsis lines were planted for 2 months.”

Reviewer 2 Report

· The authors did a good job of understanding the secondary cell wall synthesis mechanisms in Sudan grass by transcriptomic analysis followed by the generation of overexpression plants.

·   Start the abstract with a clear statement on the scope, relevance, and intention of the study before describing the main results. End the abstract with a clear statement about the main conclusions and perspectives of the work.

·   Many sentences do not make sense and should be clearly rewritten. Authors are urged to go through the text and correct it. The quality of the english language and grammar for the entire manuscript needs to improve by a native English speaker.

· If possible, restructure and carefully edit the results and discussion sections and add clear information regarding the most noteworthy findings. The author should mainly discuss more how similar/dissimilar this study's result is with previous studies.

·  Detail the number of plants/ replications used for transcriptome analysis. It is useful to judge the quality of the presented data.

·     Provide high-resolution images for figures 3 and 4. The font size is not visible.

· There is no detailed formation about over-expression plants, for instance, which generation, and how plants are selected for each generation?

Author Response

Dear reviewer,

We appreciate the critical comments and useful suggestions from you. All the questions and comments have been acknowledged. The manuscript has been also rewritten and organized as suggestions. Herein we re-submit our manuscript entitled “Sudan Grass Transcriptome Analysis Identified CcNAC1 Gene to Function in Secondary Cell Wall Formation” (Manuscript ID: ijms- 2228977) for consideration as article by International Journal of Molecular Sciences.

Sincerely,

Dr. Xiaoxian Zhong

Reviewer: Comments on the manuscript " Sudan Grass Transcriptome Analysis Identified CcNAC1 Gene to Function in Secondary Cell Wall Formation ", by Huang et al..

  1. Start the abstract with a clear statement on the scope, relevance, and intention of the study before describing the main results. End the abstract with a clear statement about the main conclusions and perspectives of the work.

Re: Thank you. We have modified abstract according to your suggestion.

Sudan grass is a high-quality forage of sorghum. The degree of lignification of Sudan grass is the main factor affecting the digestibility of ruminants such as cattle and sheep. Almost all lignocellulose in Sudan grass is stored in the secondary cell wall, but the mechanism of synthesis of the secondary cell wall in Sudan grass is still unclear. In order to study the mechanism of secondary cell wall synthesis in Sudan grass, this study used the in vitro induction system of Sudan grass secondary cell wall established earlier, and found that the NAC transcription factor CcNAC1 gene was related to the synthesis of Sudan grass secondary cell wall through transcriptome sequencing. In order to study the function of CcNAC1 gene in secondary cell wall synthesis, this study generated CcNAC1 overexpression lines, and found that overexpression of CcNAC1 can significantly increase the lignin content in plants. Through subcellular localization analysis, it was found that CcNAC1 could be expressed in the plant nucleus. In addition, we used yeast two-hybrid screening to find 26 interacting proteins with CcNAC1, and GO and KEGG analysis found that CcNAC1 was related to metabolic pathways, and biosynthesis of secondary metabolites. In summary, the synthesis of secondary cell wall of Sudan grass can be regulated by CcNAC1.”

  1. Many sentences do not make sense and should be clearly rewritten. Authors are urged to go through the text and correct it. The quality of the english language and grammar for the entire manuscript needs to improve by a native English speaker.

Re: The manuscript was revised and re-edited by Augustine Antwi Boasiako, a native English speaker author from Ghana.

  1. If possible, restructure and carefully edit the results and discussion sections and add clear information regarding the most noteworthy findings. The author should mainly discuss more how similar/dissimilar this study's result is with previous studies.

Re: Thank you. We have re-edited discussion section according to your suggestion.

  1. Detail the number of plants/ replications used for transcriptome analysis. It is useful to judge the quality of the presented data.

Re: In this study, the transcriptome sampling was untreated Sudan grass callus as control, and Sudan grass callus treated with TEM+Light18day, TEM+Dark18day, TEMB2C5+Light18day, and TEMB2C5+Dark18day as treatment samples. Each transcriptome sample was scraped from 50 superficial cells of Sudan grass callus with normal appearance observed under a microscope, wrapped in aluminium-foil, quick-frozen in liquid nitrogen, and stored at minus -80℃. Transcriptome sequencing was performed on one control sample and four treated samples. After the analysis of the transcriptome data, perform qRT-PCR verification on the genes of interest and the genes that are significantly up-regulated and down-regulated. Three biological replicates were performed for each gene, and three technical replicates were performed for each biological replicate to ensure the accuracy of the data.

  1. Provide high-resolution images for figures 3 and 4. The font size is not visible.

Re: We had uploaded the original figures when we submitted.

  1. There is no detailed formation about over-expression plants, for instance, which generation, and how plants are selected for each generation?

Re: Thank you. We have added specific Arabidopsis transformation and screening methods.

The full-length CDS of the CcNAC1 gene was cloned into the pTF101.1 vector under the control of the CaMV 35S promoter. After the vector sequence was confirmed by sequencing, the recombinant pTF101.1-CcNAC1 plasmid vector was transformed into Arabidopsis thaliana cultivar Columbia-0 using Agrobacterium tumefaciens strain EHA101. The selection marker gene (bar) and a 35S promoter were amplified by PCR to verify the positive transgenic plants. Arabidopsis plants were transformed by the floral dip method. The seeds harvested after infection were T1 generation. T1 seeds were collected and selected on MS medium containing 20 mg/L Basta and then tested bar, CcNAC1 and 35S promote by PCR [53]]. The seeds harvested from positive plants of T1 generation were T2 generation, and T2 generation plants were identified as positive plants by the same screening method as T1 generation plants. If all T2 generation plants of a line were positive, the line was considered to be homozygous; if separation occurred, the line was considered to be heterozygous. The seeds harvested from T2 homozygous lines were T3 homozygous lines. T3 homozygous lines were used for further study. Samples of Arabidopsis overexpressed lines and wild type were taken to determine lignin content when each Arabidopsis lines were planted for 2 months.

Round 2

Reviewer 2 Report

Accept in present form